# Effect of 10-Week Whole-Body Vibration Training on Falls and Physical Performance in Older Adults: A Blinded, Randomized, Controlled Clinical Trial with 1-Year Follow-Up

**DOI:** 10.3390/ijerph21070866

**Published:** 2024-07-02

**Authors:** Harri Sievänen, Maarit Piirtola, Kari Tokola, Tanja Kulmala, Eerika Tiirikainen, Pekka Kannus, Juha Kiiski, Kirsti Uusi-Rasi, Saija Karinkanta

**Affiliations:** 1The UKK Institute for Health Promotion Research, FI-33500 Tampere, Finland; maarit.piirtola@ukkinstituutti.fi (M.P.); saija.karinkanta@kela.fi (S.K.); 2Department of Musculoskeletal Surgery and Diseases, Tampere University Hospital, FI-33520 Tampere, Finland

**Keywords:** exercise, fall risk, fractures, muscle strength, physical functioning, prevention

## Abstract

Whole-body vibration training (WBV) training has shown positive effects on bone strength, muscle strength, and balance, but the evidence on fall prevention is not yet persuasive. This study aimed to evaluate the effectiveness of WBV training in preventing falls and improving physical performance among older adults at fall risk. The study was an assessor- and participant-blinded, randomized, and controlled 10-week training trial with a 10-month follow-up. One hundred and thirty older adults (mean age 78.5 years, 75% women) were randomly allocated into the WBV group (*n* = 68) and the low-intensity wellness group (*n* = 62). Falls were prospectively collected using monthly returned and verified diaries. Physical performance was evaluated at baseline before randomization, after the intervention, and follow-up with established methods. The data were analyzed on an intention-to-treat basis. Negative binomial regression was used to estimate the incidence rate ratios for falls, and Cox regression models were used to calculate the hazard ratios for fallers. Between-group differences in physical performance were estimated by generalized linear mixed models. The retention rate was 93%, and the mean adherence to the WBV training was 88% and 86% to the wellness training. Sixty-eight participants fell at least once, and there were 156 falls in total. In the WBV group, the incidence rate of falls was 1.5 (95% confidence interval 0.9 to 2.5) compared to the wellness group (*p* = 0.11). The hazard ratio for fallers in the WBV group was 1.29 (0.78 to 2.15) (*p* = 0.32). There was no between-group difference in physical performance after the training period, but by the end of the follow-up, WBV-related benefits appeared. The chair-rising capacity was maintained in the WBV group, while the benefit disappeared in the wellness group (*p* = 0.004). Also, the 0.5-point difference in short physical performance battery (SPPB) score favored WBV training (*p* = 0.009). In conclusion, progressive side-alternating WBV training was feasible and well-tolerated among fall-prone older adults. During the one-year follow-up, WBV training was associated with improved physical performance but did not prevent falls compared to chair-based group exercises.

## 1. Introduction

Falls contribute largely to the morbidity and mortality of older adults [1]. As the number of aging people is increasing [2], fall-induced deaths and injurious falls will increase, despite indications of flattened incidence [3,4]. This imminent scenario calls for effective actions to identify those older adults at elevated risk of falling, improve their physical functioning, and reduce their fall risk. Recently recommended actions include assessing gait and balance, reviewing polypharmacy, health status, and home and environmental hazards of falling and managing discovered risks, as well as promoting strength and balance training [5].

Of many alternatives of physical training, whole-body vibration (WBV) training has been long considered particularly compelling due to its many anticipated benefits in bone strength, muscle strength, balance, pain, and other health outcomes [6], all opportune concerns of ageing societies. Within minutes, WBV training can generate thousands of feet-mediated mechanical stimuli to the body, comparable to load magnitudes induced by walking or even jogging, but in a much shorter time [7]. Since WBV training does not particularly strain the cardiovascular system while stimulating the musculoskeletal system [8], its potential to improve physical performance and prevent falls even among individuals with a low physical capacity seems promising.

During the last two decades, tens of randomized controlled trials (RCTs) have evaluated the effects of WBV training on physical functioning and performance, body composition, bone mineral density (BMD), and prevention of falls in older adults. A systematic review and meta-analysis of 20 RCTs of adults older than 60 years by Orr showed that WBV training complemented by other exercises can improve body balance, but the effectiveness of WBV-only training remained inconsistent [9]. A systematic review and meta-analysis of eight RCTs of postmenopausal women by Ma et al. showed that WBV training alone did not improve BMD; but, if only high-quality trials were considered, then a small effect on lumbar spine BMD was evident, whereas the data on falls were deemed insufficient for appropriate data synthesis [10]. A systematic review and meta-analysis of 14 RCTs of adults older than 50 years by Jepsen et al. found no overall effect on bone variables, but the data on falls from six studies indicated a significant 33% reduction in fall rate [11]. A systematic review of 33 RCTs of adults older than 65 years by Rogan et al. showed that WBV training can improve static balance in independently living older adults and dynamic balance in older adults with functional limitations or deficits [12]. A systematic review and network meta-analysis of 30 RCTs by Lai et al., including 5 WBV trials of adults older than 60 years, found that resistance training of different modalities was most effective for muscle strength and physical performance; WBV training seemed also to be beneficial for physical performance [13]. A systematic review and meta-analysis of 10 RCTs of postmenopausal women by Marín-Cascales et al. showed that WBV training can slightly improve lumbar spine BMD; in postmenopausal women young than 65 years, it can also improve femoral neck BMD [14]. A systematic review and meta-analysis of 46 RCTs of all-aged people, including 16 trials of the elderly, by Fischer et al. found that WBV training can improve balance and gait speed among the elderly [15]. A systematic review and meta-analysis of 10 RCTs of institutionalized people older than 65 years by Alvarez-Barbosa et al. showed that WBV training can improve the functional mobility of these people [16]. A systematic review and meta-analysis of seven RCTs of older adults with sarcopenia by Wu et al. showed that WBV training can improve muscle strength and physical performance [17]. Similarly, a systematic review and meta-analysis of 18 RCTs of adults older than 65 years by Tan et al. showed WBV-training-induced improvements in muscle strength and physical performance [18]. A recent network meta-analysis of 25 RCTs by Liu et al. suggested that high-frequency WBV training may be beneficial for functional balance in older adults [19]. To summarize the results of meta-analyses, the most convincing evidence for the benefits of WBV training exists for balance, muscle strength, and physical functioning [9,12,13,15,16,17,18,19], whereas the evidence for fall prevention is not yet sufficient and calls for further high-quality trials [10,11].

At present, there is a gap in scientific knowledge when it comes to the proven effectiveness of WBV training in preventing falls among older adults. The number of pertinent randomized controlled studies is yet too small to demonstrate this. The present study adds to this knowledge by investigating whether WBV training prevents falls and improves physical functioning among older adults living in or using the services of senior centers. We hypothesized that improved physical performance achieved through progressive WBV training would translate into fewer falls among people at elevated risk of falling.

## 2. Materials and Methods

### 2.1. Study Design

The study was a multicenter RCT that comprised a 10-week training intervention at the beginning and a subsequent follow-up so that a one-year period was covered. The training intervention comprised either supervised WBV or wellness training sessions delivered for ten consecutive weeks (details of the training programs are given in Section 2.5). During two successive years, the interventions and follow-ups were conducted in four senior centers located in the city of Tampere, southern Finland. In a center-wise manner, the interventions were started either in October/November or February/March.

The assessors of the outcome variables were blinded to the group allocation, and they were not involved in supervising the training sessions.

Participants were informed that two different training programs were compared in the study, while they were kept unaware of the primary study hypothesis. In terms of fall prevention, WBV training was considered to be effective, whereas wellness training was considered to be ineffective; thus, it was deemed to be appropriate for sham training because of its low-intensity contents.

The study protocol has been registered on ClinicalTrials.gov (NCT01523600). This report is written according to CONSORT reporting guidelines [20]. The CONSORT diagram of the trial is shown in Figure 1.

### 2.2. Sample Size

The intended total sample size of participants at baseline was 200, with 100 per group. Power estimation indicated that the intended sample size, while permitting 15 dropouts/group, provided an 80% statistical power to detect a 2–5% training effect on physical performance at significance level of *p* < 0.05, should such a between-group effect exist. Since half of the participants were expected to fall annually, the sample size was considered sufficient to reveal a ~30% reduction in the incidence of falls. This relative reduction in falls was considered credible in terms of what has been reported in the literature [11].

### 2.3. Participants

Participants were mainly recruited at information events arranged at the collaborating senior centers. From those showing interest in the study, physicians screened potentially eligible participants. The medical screening examination focused on the participants’ medical and surgical histories, difficulties in daily activities, lifestyles (physical activity, nutrition, use of alcohol, and smoking), and health concerns that might impede physical training.

Inclusion criteria were being a volunteer, at least 65 years old, ambulatory with or without a walking aid, living in or using the services (accommodation, therapy, meal services, exercise, or other groups) of the collaborating senior centers, and not suffering from moderately severe or severe dementia. Exclusion criteria were coexisting conditions or previous injuries that were considered contraindications to WBV training: knee or hip prosthesis, acute musculoskeletal inflammation, severe heart disease, cardiac pacemaker, high risk of thromboembolism, recent fracture (<1 year for lower extremities or spine, and <6 months for upper extremities), recent major surgical operation, tumor, diabetic neuropathy, gall or bladder stones, hernia, or severe vertigo.

Of the 192 persons initially interested, 15 lost interest and 47 did not meet the selection criteria, leaving 130 eligible participants. Each participant gave written informed consent.

### 2.4. Randomization

After the baseline measurements, the participants were randomly allocated into WBV training (*n* = 68) or wellness training (*n* = 62) groups in stratified blocks. The baseline short physical performance battery (SPPB) score (cut-point < 8) and sex were used for stratification. One investigator who was not involved in the baseline measurements and who did not know the participants performed the stratified randomization for each center using a web tool (www.graphpad.com/quickcalcs/randomize1/, accessed on 6 June 2024).

### 2.5. Training Interventions

WBV training was individually arranged, and the participants trained twice a week for 10 weeks according to a preplanned protocol (Table 1). The safety and feasibility of this training program were verified earlier among the nursing home residents [21]. WBV training was performed on a side-alternating vibration platform (Galileo Med M Plus, Novotec GmbH, Pforzheim, Germany) under the supervision of a physiotherapist. The WBV training protocol consisted of alternate low-frequency (12 Hz), intermediate-frequency (18 Hz), and high-frequency (26 Hz) vibrations delivered in one-minute bouts separated by one-minute breaks. When standing on the vibrating platform, the participants were instructed to keep their knees slightly bent (140–160°) and carry out slight squatting, toe raises, and lateral weight transfers at their preferred pace. If a participant perceived the 26 Hz vibration frequency to be too intense, then 18 Hz was used. Progressive training was achieved by increasing the vibration amplitude by a wider foot distance and more bouts. The measured peak acceleration of the vibration platform loaded by 60 kg and 100 kg persons varied between ~1 and 2 Gs (1 G corresponds to Earth’s gravity in terms of magnitude) at 12 Hz, between ~2 and 5 Gs at 18 Hz, and between ~4 and 10 Gs at 26 Hz, depending on the foot distance from the fulcrum of the platform.

The wellness training was arranged in small groups of up to 10 participants, who attended supervised 45 min training sessions once a week for 10 weeks. The training sessions comprised mainly low-intensity chair-based stretching and flexibility exercises, supported by music and various equipment (sticks, exercise bands, and balls) to create variation in training and maintain participants’ interest. In the first two sessions, cognitive tasks were performed along with light physical training. Wellness training was deliberately designed not to include specific exercises targeted to improve balance or muscle strength. Therefore, it was initially regarded as sham training in terms of fall prevention.

Center-wise, the wellness training sessions were supervised by the same physiotherapist who supervised WBV training. The supervisors recorded both the attendance in training sessions and deviations from the intended training program in the participant’s training diary.

After the intervention period, participants were asked to evaluate the perceived effectiveness of their training on muscle strength, balance, walking ability, and vitality, as well as to evaluate comfort, safety, and willingness to continue the same type of training. An ad hoc questionnaire based on the 5-point Likert scale (1 = fully disagree, 2 = somewhat disagree, 3 = neutral, 4 = somewhat agree, 5 = fully agree) was used.

### 2.6. Outcome Variables

In addition to the 12-month prospective collection of falls, participants’ physical functioning and fear of falling were assessed at baseline before randomization, after the 10-week training intervention, and at the end of follow-up. Body height and weight were measured at baseline. Participants’ habitual physical activity was measured during the first and last intervention weeks and the final follow-up week.

The primary outcome was the number of falls. A fall was defined as “an unexpected event in which the participant comes to rest on the ground, floor or lower level” [22]. Falls were prospectively collected using fall diaries which the participants were asked to fill in daily and return monthly in prepaid envelopes. Each reported fall was verified by a telephone call. Both minor injuries (pain, bruises, muscle, or joint injuries) and moderate-to-serious injuries (fractures and head injuries) were recorded.

Physical functioning was assessed by SPPB, which comprises a static balance test, a 5-time chair-stand, and a 4 m normal gait speed tests; this provides an aggregate score (range 0–12) of lower extremity functioning [23]. The Timed Up and Go-test (TUG) was also performed [24].

Fear of falling in different activities of daily living was assessed using Falls Efficacy Scale International [25], which is a 16-item self-report questionnaire based on a 4-point scale (1 = not at all concerned to 4 = very concerned). If the respondents indicated that they did or could not perform the activity, a hypothetical response was asked.

Mean daily total physical activity time was evaluated with a hip-worn triaxial accelerometer (Hookie AM20, Traxmeet Ltd., Espoo, Finland). All movements inducing a higher than 0.0195G mean amplitude deviation of the resultant acceleration were accumulated regardless of the continuous duration of movement [26]. More than 10 h daily wear-time from four days was required for eligible data.

### 2.7. Statistical Analysis

Mean, standard deviation (SD), and 95% confidence interval (95% CI) are given as descriptive statistics. 

All prospectively collected data were analyzed on an intention-to-treat basis. Between-group differences in physical performance were estimated by linear mixed models for normally distributed outcomes and generalized linear mixed models with gamma distribution and log-link function for non-normally distributed outcomes; age, height, weight, and sex were used as covariates, and the senior center was used as a random factor.

Fall incidence rates were calculated as the total number of falls divided by the time over which the falls were monitored (in person-years) in both training groups. Negative binomial regression was used to estimate the incidence rate ratios for falls. Cox regression models were used to calculate hazard ratios for fallers in both groups.

Statistical analyses were performed with IBM SPSS Statistics for Windows, version 27.0 (IBM Corp., Armonk, NY, USA). A *p*-value < 0.05 was considered to be statistically significant.

## 3. Results

### 3.1. Participant Characteristics 

The participant characteristics in WBV and wellness groups at baseline are shown in Table 2. Of the 130 participants, 98 (75%) were women and 32 (25%) were men. Their mean age was 78.5 (SD 7.2) years. The mean SPPB score was slightly above 9, indicating at least some problems in physical functioning on average; individual scores ranged from 2 to 12. Forty-seven participants (36%) used some walking aid at least occasionally. On an average day, participants performed at least some physical activity for 2.5 h; the individual total daily times ranged from 10 min to 5.5 h. On average, participants suffered from almost four chronic diseases. Only three participants (2%) reported no chronic diseases. Most participants (63%) lived independently in their own homes, 25% lived independently in senior houses, and 12% needed sheltered housing. 

### 3.2. Retention Rate and Adherence

Seven (5%) participants withdrew before starting the training, one during the third training week, and one person died during the last month of the follow-up (Figure 1). Altogether, the overall retention rate was 93%. Among those 120 participants (92%) who attended at least one training session, the mean adherence to training was 88% in the WBV group and 86% in the wellness group. Center-wise, adherence varied between 84% and 91% in the WBV groups and 80% and 92% in the wellness groups. Nine participants in the WBV group (14%) discontinued training with 26 Hz once having tried it but continued with 18 Hz; two of these persons ceased training later.

### 3.3. Adverse Events

Both training types were well-tolerated without any serious adverse incidents. Few participants reported temporary muscle soreness, joint pain, or dizziness. In the WBV group, seven participants (11% of trainees) ceased training because of incident health issues, whereas none in the wellness group did so once having started the training. Two health issues were not related to WBV training, whereas five pertained to back, hip, or knee pain alone or together with a feeling of weakness or sickness, or continuous perceived dizziness during training. All these issues were resolved without the need for medical treatment.

### 3.4. Perceived Training Benefits

Out of 122 participants, 113 (93%) responded to the questionnaire concerning their perceptions of training they had performed. The mean scores in the WBV and wellness groups were 3.5 and 3.6 for muscle strength (*p* = 0.42), 3.4 and 3.5 for balance (*p* = 0.64), 3.4 and 3.6 for walking ability (*p* = 0.55), 3.4 and 4.0 for vitality (*p* = 0.004), 4.5 and 4.8 for comfort (*p* = 0.012), 4.7 and 4.9 for safety (*p* = 0.19), and 3.8 and 4.5 for willingness to continue training (*p* = 0.045), respectively.

### 3.5. Effect on Falls

Fall diaries covered 1463 person-months (about 122 person-years). Accumulation of falls in both groups is illustrated in Figure 2. Twenty-six participants (41%) in the WBV group and twenty-eight (48%) in the wellness group saw out the one-year study period without falls. Sixty-eight participants reported a total of 156 falls, comprising 99 in the WBV group and 57 in the wellness group. Of the fallers, 33 (14 and 19) fell once, 18 (12 and 6) fell twice, and 17 (10 and 7) endured multiple falls; the number of falls in the latter group ranged from 3 to 15. Compared to the wellness group, the incidence rate of falls was 1.50 (95% CI 0.91 to 2.49) in the WBV group (*p* = 0.11). Likewise, the hazard ratio for fallers was 1.29 (0.78 to 2.15) in the WBV group (*p* = 0.32).

In the WBV group, half of the falls and 44% in the wellness group were non-injurious without reported consequences. Minor-fall-induced injuries, mainly pain and bruises, occurred in 36% of falls in the WBV group and 51% of falls in the wellness group. A total of 16 falls (10%) led to moderate-to-serious consequences: 11 head injuries occurred, along with 1 skull fracture, 1 upper limb fracture, 2 lower limb fractures, and 2 vertebral fractures. The time of occurrence of fall-induced injuries in both groups is illustrated in Figure 2. These injuries occurred mostly (69%) in the WBV group, mainly during the latter half of follow-up.

### 3.6. Effect on Physical Performance

Mean percentage changes in lower limb physical performance are summarized in Table 3. After the 10-week training intervention, there were no significant between-group or within-group differences, but by the end of the follow-up, the WBV group showed a significant 0.5-point net benefit (*p* = 0.009) in the SPPB score compared to the wellness group. Similarly, the chair-stand time in the WBV group continued decreasing and reached a significant 1.5 s benefit (*p* = 0.004) compared to the wellness group. In contrast, the initially slower walking speed in the wellness group reached the level of that in the WBV group (*p* = 0.026) by the end of the follow-up. The TUG test did not show a significant between-group difference (*p* = 0.86).

### 3.7. Effect on Fear of Falling and Physical Activity

There was no effect on fear of falling (*p* = 0.88) nor daily physical activity (*p* = 0.37). At follow-up, mean scores for fear of falling were only 0.3 and 0.6 points higher in the WBV and wellness groups, respectively. Also, at follow-up, the mean daily physical activity time was only one minute longer in the WBV group and four minutes shorter in the wellness group. After the training interventions, the mean physical activity times were 10 and 6 min shorter, respectively.

## 4. Discussion

The present study assessed the effectiveness of progressive high-magnitude WBV training in reducing falls, improving physical performance, reducing fear of falling, and increasing habitual physical activity among older adults who were likely to benefit from this type of training. However, compared to a low-intensity wellness (sham) training, no consistent-WBV-training-related effects were evident in any outcome variable. The participants in both groups represented truly a proper target group at elevated risk of falling, as more than half of them fell at least once during the study (see Section 3.5). The assessors were blinded to the group assignment, while the study design permitted a pseudo-blinded implementation of training. This approach was validated by showing that the participants rated both training types similarly beneficial, even slightly favoring wellness training over WBV training (see Section 3.4). The WBV training program was previously shown to be feasible and safe among frail nursing home residents [21], which likely explained the excellent 90% retention rate and adherence to the training in the present study (see Section 3.2) and a low incidence of training-related adverse events (see Section 3.3). Altogether, the strong study design and its successful execution provided a solid basis to verify the anticipated benefit of WBV training in fall prevention, should it have existed in the present study population.

Present WBV training was progressive in terms of intensity and duration of sessions in contrast to the execution of wellness training, which was intentionally delivered once a week without exercises that would primarily have improved balance or muscle strength. However, against our expectations and existing scientific evidence on the benefits of WBV training on physical performance [9,12,13,15,16,17,18,19], the effects on lower limb physical performance were similar in both groups, although some long-term effects appeared during the follow-up (see Table 3). In the WBV group, the SPPB score showed a significant 0.5-point benefit, evidently due to the maintenance of improved chair-rising capacity that disappeared in the wellness group. In the pilot study of the present trial, where the same 10-week WBV training program was employed in a small group of nursing home residents, a comparable 0.4-point improvement was observed [21]. Whereas a mean 0.5-point gain in the SPPB score can be considered a clinically meaningful improvement in lower limb physical performance [27], no positive WBV-training-related effect on the incidence of falls was found, nor were any respective trends observed (see Figure 2). 

The meta-analysis by Jepsen et al. [11] suggests that WBV training may reduce falls in older adults by one third. However, scrutinizing the original four WBV trials reporting the fall data revealed that the evidence is not as convincing as commonly believed. In their 18-month RCT of 151 postmenopausal women aged 68 years on average, von Stengel et al. observed 35 falls in the WBV group and 76 falls in the control group engaged in low-intensity wellness training, similar to the present study, indicating a beneficial effect on fall prevention [28]. Buckinx et al., in their 6-month RCT and 6-month follow-up of 62 nursing home residents aged 82 years on average, noticed 32 falls in the WBV group and 29 falls in the control group, indicating no effect on falls [29]. Leung et al., in their 18-month RCT of 710 postmenopausal women aged 73 years on average, found only 70 falls in the WBV group but 122 falls in the control group, showing a statistically significant, meaningful reduction in falls [30]. Sitja-Rabert et al., in their 6-week RCT and 6-month follow-up of 159 institutionalized older adults aged 82 years on average, noticed 33 falls in a combined WBV and exercise group and 24 falls in the control exercise group, indicating no effect on falls [31]. Combining the fall data from the above four RCTs and the present study yield 269 and 308 falls in the WBV and control groups, respectively, indicating about 13% benefit for WBV training in fall prevention. However, if the largest RCT by Leung et al. utilizing low-magnitude WBV training (peak acceleration 0.3G) was excluded [30], this benefit completely vanished. 

While several RCTs have shown that WBV training can improve physical performance in older adults, this improvement in physical functioning may not necessarily translate into fewer falls as the present findings indicated. In the above-assessed four RCTs, improved lower limb physical performance and reduced incidence of falls were reported in one trial [30] but not in the other [31]. Falling is a multifactorial event that depends on several intrinsic and extrinsic factors [5,32], with these sometimes being just a consequence of unfortunate conditions and thus virtually unpreventable.

An adequate follow-up after an intervention study is important for verifying the maintenance of achieved training effects. In the present study, however, no significant between-group differences in any outcome variable were observed immediately after the 10-week intervention. Regarding the incidence of falls, the rate in the WBV group appeared somewhat higher from the beginning of the study compared to the wellness group, and most (about 80%) fall-related injuries occurred after the 10-week intervention period (see Figure 2). By the end of the follow-up, the SPPB score was increased in the WBV group compared to the wellness group, while the initially somewhat lower chair-rising capacity in the WBV group reached the level of the wellness group. In the wellness group, in turn, the initially somewhat lower normal walking speed reached the level of the WBV group. Whether the observed changes in physical performance during the follow-up were attributable to specific training interventions remains speculative only.

Prevention of falls among frail persons with WBV training might be effective, but recruitment of these people into an intervention trial proved challenging. One of our intended initial inclusion criteria was that the eligible participant should have at least some difficulties in mobility or physical functioning. However, we had to abandon this criterion at the beginning of enrollment (in the first senior center) because of a low recruitment rate and a paucity of eligible participants. Therefore, the present participants represented a heterogeneous population from institutionalized, frail persons with poor physical functioning to healthy home-dwelling persons with excellent physical performance. Notwithstanding that WBV training may not be more efficient than conventional physical training [13], at best, it might offer a fast and intriguing option for effective physical rehabilitation of older adults and some variability to conventional physical exercises. Although we initially presumed wellness training ineffective, to our surprise, this was not the case. Interestingly, a recent meta-analysis discovered the utility of chair-based exercises for older adults’ physical functioning [33].

The strengths of the present study include blinded randomized controlled design of the training intervention with a long follow-up, a relatively large target group at elevated risk of falling, pretested WBV training program, high retention rate, excellent adherence to both types of training, prospective collection of falls with verified diaries, prospective assessment of physical performance and fear of falling with established methods, collection of device-measured daily physical activity, and conducting the WBV intervention as per recommendations [34]. Further, the design of the present WBV study complied with the pilot study [21], which received high PEDro scores on methodological quality in earlier meta-analyses [9,16].

The lack of an inactive non-exercising control group may be considered a limitation of the present study. This preplanned choice of study design prevented us from comparing whether WBV and wellness training programs would have been effective compared to being inactive, not only whether their effectiveness would have differed, which did not happen. Also, the training period could have been longer than 10 weeks, but most likely at the price of a reduced retention rate and adherence to training. Further, because of many alleged contraindications to WBV training, many potentially eligible older adults needed to be excluded, reducing the generalizability of present findings. Some older adults with such contraindications, like knee prostheses or low BMD, might have benefited from WBV training without additional health risks [35,36]. Although safety should always come first, too much precaution may not be necessary. We, therefore, encourage broader reporting of clinical experiences with WBV training in rehabilitating older adults with different health conditions and diseases.

Ultimately, the study needed to be terminated because of depleted funding. Consequently, the planned training interventions and follow-up could not be executed in two more senior centers. This led to a smaller sample size than the intended 200 participants, meant to be recruited at the beginning of the trial. Nevertheless, the present trial is among the largest WBV trials. Regarding the prevention of falls, the main outcome of the present study, the result would hardly have been inverted in a larger sample of similar older adults. The importance of the present study lies thus in the fact that its main result challenges the common belief that WBV training can prevent falls among older adults. Further high-quality RCTs are warranted.

## 5. Conclusions

A progressive 10-week high-magnitude WBV training program performed twice a week was feasible and well-tolerated among fall-prone older adults but proved ineffective in preventing falls compared to chair-based group exercises performed once a week. During the one-year follow-up, WBV training was associated with improved lower limb physical performance. Although WBV training may not be more efficient than conventional physical training, it yet offers a noteworthy option for training and rehabilitation of older adults irrespective of their level of physical functioning. There is a need for adequately powered and long training intervention trials to confirm the anticipated health benefits of WBV training and to determine appropriate specifications for effective WBV training in relevant target groups.

## Figures and Tables

**Figure 1 ijerph-21-00866-f001:**
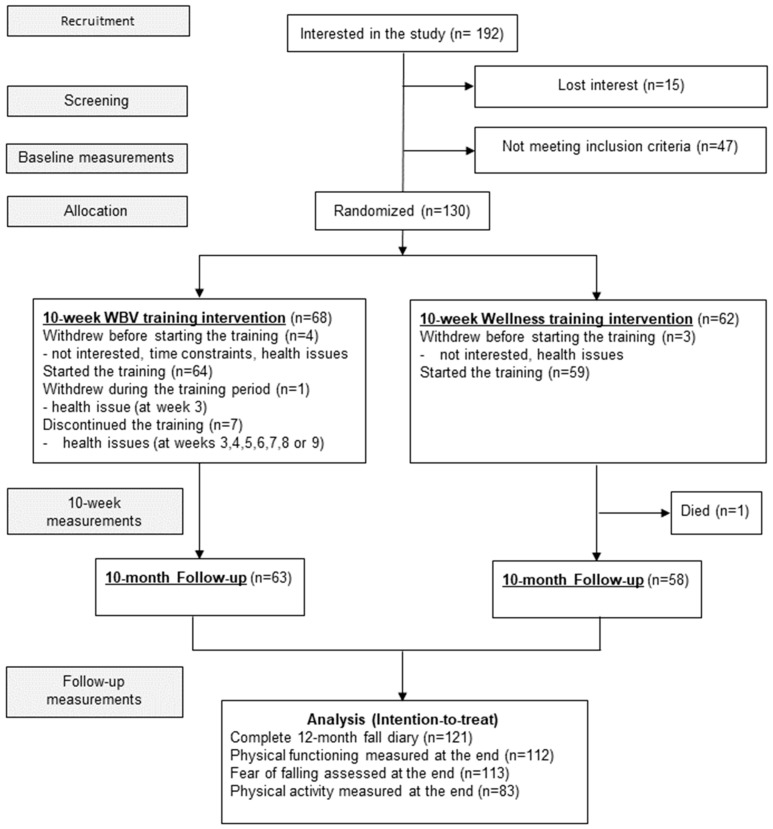
The CONSORT diagram of the trial.

**Figure 2 ijerph-21-00866-f002:**
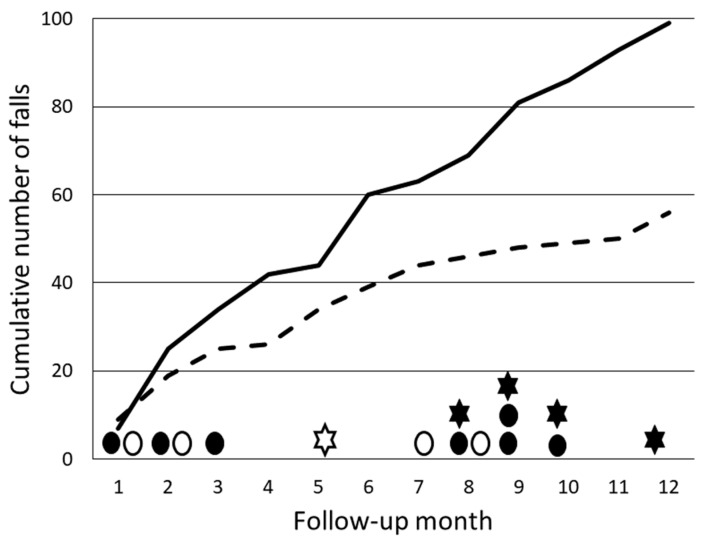
Accumulation of falls in the WBV group (solid line) and wellness group (dashed line). The time of occurrence of bone fractures (star) and head injuries (circle) in these groups is indicated by solid and open symbols, respectively.

**Table 1 ijerph-21-00866-t001:** Ten-week WBV training program comprising 20 training sessions in total and providing 80 min effective vibration training time during the ten-week intervention.

Week	No. of Sessions	Duration * (min)	Vibration Frequency (Hz)	Vibration Amplitude (mm Peak-to-Peak)	Additional Exercises † in Weekly Sessions
1	2	1 × 1	12	~2	A; A
2	2	2 × 1	12, 18	~2–4	B/A; A/B
3	2	3 × 1	12, 18, 12	~4	B/A/B; A/B/A
4	2	4 × 1	12, 18, 12, 26	~4–6	B/A/B/A; A/B/A/B
5	2	5 × 1	12, 26, 12, 26,12	~4–6	B/A/B/A/C; A/B/A/B/C
6	2	5 × 1	12, 26, 12, 26,12	~6	C/A/C/A/B; A/C/A/C/B
7	2	5 × 1	26, 12, 26,12, 26	~6	A/B/C/B/A; B/A/C/A/B
8	2	5 × 1	12, 26, 12, 26,12	~6–8	B/C/B/C/B; C/B/C/B/C
9	2	5 × 1	26, 12, 26,12, 26	~6–8	A/C/B/A/C; C/B/A/C/B
10	2	5 × 1	26, 12, 26,12, 26	~6–8	B/A/C/B/C; C/B/A/C/B

* One-minute break between the training bouts; † A = slight squatting, B = toe raises, C = lateral weight transfers; all exercises were performed at each participant’s preferred pace.

**Table 2 ijerph-21-00866-t002:** Baseline clinical characteristics (mean, SD) of the study groups.

Clinical Characteristics	WBV Group *n* = 68	Wellness Group *n* = 62
Male/female (*n*)	18/50	14/48
Age (years)	77.8 (6.8)	79.2 (7.6)
Height (cm)	162.4 (8.6)	162.7 (8.8)
Weight (kg)	72.0 (11.9)	74.3 (13.3)
Perceived health score (range 1–5, 5 best)	2.7 (0.7)	2.7 (0.7)
Physical functioning		
SPPB score (range 0–12, 12 best)	9.2 (2.7)	9.3 (2.6)
Balance score (range 0–4, 4 best)	3.3 (1.2)	3.2 (1.1)
Normal walking speed (m/s)	0.96 (0.57)	0.87 (0.27)
Chair-stand time (s)	15.4 (5.9)	13.9 (4.2)
TUG-test time (s)	11.6 (8.0)	10.8 (7.1)
Fear of falling score (range 16–64, 64 highest fear)	26.4 (11.2)	23.5 (5.6)
Perceived capacity in daily living score (range 1–5, 5 best)	2.4 (0.8)	2.2 (0.9)
Daily physical activity time (min)	150 (71), *n* = 54	149 (70), *n* = 51
Use of walking aids at least occasionally		
Cane	10 (15%)	10 (16%)
Walker	10 (15%)	11 (18%)
Wheelchair	0 (0%)	1 (2%)
Crutch	3 (4%)	2 (3%)
Reported common chronic diseases and conditions		
Number of diseases	3.7 (1.9)	3.7 (1.9)
Number of medications	4.7 (3.3)	6.2 (3.8)
Hypertension	36 (53%)	34 (55%)
Osteoarthritis	31 (46%)	33 (53%)
Cardiovascular diseases	29 (43%)	34 (55%)
Eye problems	31 (46%)	32 (52%)
Back problems	25 (37%)	23 (37%)
Hypothyroidism	16 (24%)	15 (24%)
Diabetes	12 (18%)	15 (24%)
Osteoporosis	10 (15%)	10 (16%)

**Table 3 ijerph-21-00866-t003:** Mean percentage changes * (95% CI) in lower limb physical performance in WBV and wellness training groups as per an intention-to-treat analysis.

Outcome	%-Change after the 10-Week Intervention	%-Change at the End of One-Year Follow-Up	Between-Group Difference
SPPB score			
WBV group	2.9 (−1.8 to 7.7)	4.3 (−0.6 to 9.3)	*p* = 0.009
Wellness group	1.0 (−3.9 to 5.9)	−1.0 (−6.1 to 4.0)	
Normal walking speed			
WBV group	−0.1 (−7.3 to 7.0)	−2.6 (−9.7 to 4.4)	*p* = 0.026
Wellness group	6.6 (−1.6 to 14.7)	8.0 (0.0 to 16.0)	
Chair stand time			
WBV group	−9.5 (−16.0 to −3.1)	−10.5 (−17.8 to −3.1)	*p* = 0.004
Wellness group	−7.4 (−14.8 to 0.1)	−2.4 (−10.9 to 6.0)	
Timed Up and Go—test time			
WBV group	−1.3 (−14.1 to 11.6)	1.0 (−4.9 to 7.0)	*p* = 0.86
Wellness group	11.0 (−3.7 to 25.7)	3.9 (−2.9 to 10.6)	

* Adjusted for age, height, weight, center, and sex.

## Data Availability

The participants of this study did not give written consent for their data to be shared publicly, so data are not made available due to privacy and ethical issues.

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
