# Peer review of "Effect of 10-Week Whole-Body Vibration Training on Falls and Physical Performance in Older Adults: A Blinded, Randomized, Controlled Clinical Trial with 1-Year Follow-Up"

_ijerph, 2024, doi:10.3390/ijerph21070866_

Round 1

Reviewer 1 Report

Comments and Suggestions for Authors

I believe that the analyzed scientific work fully respects the specific methodology of a scientific work. However, please complete the bibliographic sources because I believe that only 36 titles can provide the theoretical basis of your study.

Also, updating the bibliography with more recent titles (2021-2024) would be indicated.

Author Response

Comment 1. I believe that the analyzed scientific work fully respects the specific methodology of a scientific work. However, please complete the bibliographic sources because I believe that only 36 titles can provide the theoretical basis of your study.

Also, updating the bibliography with more recent titles (2021-2024) would be indicated.

Response: First, your overall opinion on our study is greatly appreciated.

Thank you for this relevant question. First, regarding the total number of 36 references in our paper, among many scientific journals, there is a pressure of limiting the number of references only to the most relevant ones. Accordingly, our strategy in the literature search was to find out the existing scientific evidence for the effectiveness of WBV training on falls prevention among older adults. We therefore focused on meta-analyses concerning the effects of WBV training in the target group of interest (Refs 9 -19) and the few randomized controlled WBV interventions that have so far presented data on actual falls (Refs 28 -31).

In fact, we recently updated the literature search and found only two newer relevant meta-analyses by Tan et al. and Liu et al. (Refs 18 and 19) and no newer WBV RCTs showing results on falls. The present RCT adds to the gap on this information. The lack of newer relevant studies was surprising to us as well, but to the best of our knowledge, we believe that the cited literature in our paper complies with the state-of-the-art on this current topic. Hopefully you can agree with us.

Comment 2. The authors must be commended for carrying out a study on the effect of whole-body vibration training on falls and physical performance in older adults This topic is very interesting and important. The research methodology used in the study is appropriate, and the manuscript is written with great clarity. However, some issues need to be taken into consideration.

Response: We greatly embrace your overall opinion of our study. Thank you very much.

Reviewer 2 Report

Comments and Suggestions for Authors

The authors must be commended for carrying out a study on the effect of whole-body vibration training on falls and physical performance in older adults This topic is very interesting and important. The research methodology used in the study is appropriate, and the manuscript is written with great clarity. However, some issues need to be taken into consideration.

Abstract

Please emphasize the statistical analysis used in the study.

Introduction

Please add one or a few sentences at the beginning of the last paragraph where you will present the lack of previous similar studies and thus justify the need for the present study.

I suggest adding one sentence at the end of the discussion section where you will emphasize the importance of the present study.

Methods

2.1. Study design: It is not quite clear, did you implement the training program for 10 weeks in a row, or you arrange it throughout the year? In both cases, please elaborate on the reason for that kind of study design.

2.2. Participants: Please emphasize a place in the results section where readers can see the participants' characteristics (Table 2).

2.4. Training interventions: Regarding the wellness training intervention, were you guided by some previous similar studies?

2.4. Training interventions, last paragraph: Did you implement some well-known questionnaire?

Results

I suggest moving Figure 1 to the Methods section.

Discussion

When referring to the study results, I suggest emphasizing a place in the results section where the reader can see the mentioned (e.g. Table 3)

Generally, congrats on the discussion section.

Author Response

Comment 1. Abstract

Please emphasize the statistical analysis used in the study.

Response: Thank you. Statistical methods are now added to the abstract (lines 18 – 21)

 Comment 2. Introduction

Please add one or a few sentences at the beginning of the last paragraph where you will present the lack of previous similar studies and thus justify the need for the present study.

Response: A highly relevant point, thank you. We have now added new text to the beginning of the last paragraph of introduction (lines 89 – 92).

Comment 3. I suggest adding one sentence at the end of the discussion section where you will emphasize the importance of the present study.

Response: Again, a good point. A new sentence has been added to the paragraph in the discussion before the conclusion (lines 428 - 430).

 Comment 4. Methods

2.1. Study design: It is not quite clear, did you implement the training program for 10 weeks in a row, or you arrange it throughout the year? In both cases, please elaborate on the reason for that kind of study design.

Response: We have now clarified the timing and duration of the training intervention (lines 99 – 104). Arguments for the chosen study design are given on lines 109 – 111, and further information was already presented in previous Section 2.4, now Section 2.5 (lines 176 – 183).

The basic idea was to devise a research protocol where both groups receive an intervention (of which the other is considered effective but not revealed to the participants). This success of this was verified by asking the participants’ perceptions after the training intervention.

Comment 5. 2.2. Participants: Please emphasize a place in the results section where readers can see the participants' characteristics (Table 2).

Response: The text has been rephrased to make this clear (lines 234 – 235). Also, Table 2 is now relocated.

Comment 6. 2.4. Training interventions: Regarding the wellness training intervention, were you guided by some previous similar studies?

A good point. Four of our research team (SK, MP, TK, and ET) have background in physiotherapy and physical rehabilitation, and two of them (SK and MP) did their doctoral dissertations on pertinent topics (falls and fracture prevention among older adults).  So, our intention was to devise a sham training program based on low-intensity training that would be well-adopted by older adults but be basically ineffective in terms of physical performance or falls prevention. The results of the questionnaire presented in Section 3.4 confirmed this assumption. The term “wellness training” has also been used for the same type of low-intensity training by von Stengel et al (Ref 28), now mentioned on line 352 - 353.  

Comment 7. 2.4. Training interventions, last paragraph: Did you implement some well-known questionnaire?

Response: We considered this information on participants’ perceptions about their training relevant for assessing the success of blinding the participants. For this purpose, we devised and used an ad-hoc questionnaire. We have rephrased the text accordingly (lines 191 -192).

 Comment 8. Results

I suggest moving Figure 1 to the Methods section.

Response: Figure 1 is now moved to the Methods section and referred to in Section 2.1. (lines 113 – 114).

 Comment 9. Discussion

When referring to the study results, I suggest emphasizing a place in the results section where the reader can see the mentioned (e.g. Table 3)

Response: Thank you for this comment. We have revised Discussion and now referred to respective sections, figures and/or tables in the Results, as appropriate.

Comment 10. Generally, congrats on the discussion section.

Response: Your overall opinion is much embraced. Thank you very much.

Reviewer 3 Report

Comments and Suggestions for Authors

The study is titled "Effect of whole-body vibration training on falls and physical performance in older adults: a blinded, randomized controlled clinical trial with one-year follow-up".

In the summary it is necessary to identify the groups in the sections referring to the methodology.

Introduction

The introduction is well written, however the section from line 49-80 is very long and contains a lot of detailed information that does not fit for an introduction, I suggest revising it and making it more succinct.

The objective is ok.

Methodology

I didn't quite understand what the "wellness training" was, did the group do any physical exercise? It is necessary to clarify further.

Randomization was presented, but the sample calculation was not presented. Review.

The intervention was explained in the relevant item, was the vibration training done only in those 3 exercises?

I found the sample calculation in the statistics item, check which item the magazine indicates, I think it is more appropriate to be presented in the sample.

The Retention rate and adherence item is important.

Why are the results in table 3 presented in average percentages?

I also suggest reviewing table 3 because it is showing repeated items.

Review lines 293-295, as it is not a discussion-starting text. This moment is to answer the objective of the study.

The section up to line 346 is a large summary of several studies and reviews. However, there is no discussion with the results of the present study. Review this excerpt.

I do not believe that the limitation point raised (not having an inactive group) is a limitation. The methodology was very well-designed, and I believe this point was discussed prior to the study. I suggest reviewing.

The follow-up results were not discussed, after all it was a well-designed study. These results need to be discussed.

The study is well structured, but needs some adjustments in the discussion, discussing mainly with other studies that used vibration training is essential. In addition to some adjustments to the introduction.

Comments on the Quality of English Language

 Minor editing of English.

Author Response

Comment 1. The study is titled "Effect of whole-body vibration training on falls and physical performance in older adults: a blinded, randomized controlled clinical trial with one-year follow-up".

In the summary it is necessary to identify the groups in the sections referring to the methodology.

Response: Thank you for noting this issue. We have now added more details about the training groups in the lines 15 – 16.

 Comment 2. Introduction

The introduction is well written, however the section from line 49-80 is very long and contains a lot of detailed information that does not fit for an introduction, I suggest revising it and making it more succinct.

Response: Thank you for your overall comment and for raising this concern. On one hand, we agree that bulk citations of all pertinent meta-analyses (as summarized in lines 85 – 88) would basically suffice, but on the other hand, we wanted to describe shortly the main findings and the number of RCTs evaluated in each meta-analysis so that the existing evidence-base becomes clear to the readers. We hope that you will agree with this choice.  

Comment 3. The objective is ok.

Response: Thank you very much.

Comment 4. Methodology

I didn't quite understand what the "wellness training" was, did the group do any physical exercise? It is necessary to clarify further.

Response: Wellness training comprised supervised low-intensity physical training once a week. Its were already explained in Section 2.5. (lines 176 – 183). In addition, the wellness training is addressed in Section 2.1 ((lines 110 – 111) to emphasize its low-intensity nature.

Comment 5. Randomization was presented, but the sample calculation was not presented. Review.

Response: Thank you for this relevant comment. Compared to the first submitted version of our manuscript, we have now clarified the sample size calculation and written as a new separate Section 2.2.

Comment 6. The intervention was explained in the relevant item, was the vibration training done only in those 3 exercises?

Response: As described in Section 2.5 (2.4 in the first submitted version), WBV training was performed twice a week as per the protocol described in Table 1.  The training comprised one-minute bouts of WBV training at specified frequency and amplitude during which the participants were asked to do additional exercises at their own pace according to the protocol in Table 1. Vibration was the primary training mode whereas light squatting (A), toe raises (B), and lateral weight transfers (C) served as additional exercises. Please note that for the sake of clarity we replaced the numbers (1/2/3) by letters (A/B/C) to distinguish them from meaningful concrete numbers (e.g., number of training sessions)   

Comment 7. I found the sample calculation in the statistics item, check which item the magazine indicates, I think it is more appropriate to be presented in the sample.

Response: As per your request and responded above, we have provided a separate Section 2.2 for the sample size calculation.

Comment 8. The Retention rate and adherence item is important.

Response: Quite agree. Thank you.

Comment 9. Why are the results in table 3 presented in average percentages?

Response: It is a common and comprehendible way to present changes in physical performance outcomes as mean percentage changes. Also, the power estimation was based on anticipated percentage changes.

Comment 10. I also suggest reviewing table 3 because it is showing repeated items.

Response: Thank you for noting this. For an unknown reason, the contents of Table 3 were triplicated. Extra rows have now been removed.

Comment 11. Review lines 293-295, as it is not a discussion-starting text. This moment is to answer the objective of the study.

Response: Thank you for this comment. While we wanted to reiterate the purpose of the study (the original first sentence), we slightly rephrased the first sentence and have now stated the lack of finding any consistent training effects on outcome variables in the new second sentence (lines 320 – 321). The remaining part of the first paragraph argues for the facts that made the present study strong and appropriate for the purpose and able to reveal an effect should it have existed in the present study population.

Comment 12. The section up to line 346 is a large summary of several studies and reviews. However, there is no discussion with the results of the present study. Review this excerpt.

Response: In fact, after presenting the results of few previous studies reporting the data on falls, we combined the existing previous data with the present results and ended up with a conclusion (lines 362 – 366). Please note that the reviewer 2 appreciated our way to discuss our findings.

Comment 13. I do not believe that the limitation point raised (not having an inactive group) is a limitation. The methodology was very well-designed, and I believe this point was discussed prior to the study. I suggest reviewing.

Reponse: Quite right. Your opinion is relevant that I can buy. However, it is still a common notion among researchers that the lack of a pure control group is a limitation. Upon your comment, we have slightly rephrased the limitation paragraph (starting from line 410).

Comment 14. The follow-up results were not discussed, after all it was a well-designed study. These results need to be discussed.

Response: This is a relevant point. A new paragraph addressing the follow-up data is added to the discussion (lines 375 – 386),

Comment 15. The study is well structured, but needs some adjustments in the discussion, discussing mainly with other studies that used vibration training is essential. In addition to some adjustments to the introduction.

Thank you for this overall comment. Since the scientific literature on the effects of WBV training is vast, we decided to focus on relevant meta-analyses and the few RCTs which directly evaluated the effectiveness of WBV training of falls prevention.

Comment 16. Comments on the Quality of English Language

 Minor editing of English.

Response: The text has been proofread to remove typos and some inconsistencies in language.

Round 2

Reviewer 3 Report

Comments and Suggestions for Authors

The authors took my previous comments into account.